# SARS-CoV-2 nonstructural protein 3 remodels the phosphorylation of target proteins via protein-protein interactions

Hui Yang,[1,2] Daxin Peng,[2] Luis Martinez-Sobrido,[1] Chengjin Ye[1]

**ABSTRACT** Severe acute respiratory syndrome coronavirus 2 (SARS-CoV-2), the causative agent of coronavirus disease 2019 (COVID-19), triggered a global pandemic with a significant impact on human health. The molecular basis of its pathogenicity remains incompletely understood. The viral nucleocapsid (N) protein, the most abundant protein expressed during SARS-CoV-2 infection, is thought to contribute to disease progression. Yet, its interaction network in the context of viral infection remains largely unexplored. Here, we generated a recombinant (r)SARS-CoV-2 expressing a Strep-tagged N protein by using a reverse genetics system. Affinity purification and mass spectrometry identified an interaction between SARS-CoV-2 N protein and the nonstructural protein 3 (NSP3). Domain mapping revealed that the N dimerization domain and the N-terminal region of NSP3 mediate this interaction. Notably, an N protein mutant lacking its N-terminal domain exhibited enhanced binding to NSP3 and underwent dephosphorylation, implicating NSP3 as a potential viral phosphatase. We further found that NSP3 interacts with interferon regulatory factor 3 (IRF3), a key transcription factor involved in host type I interferon (IFN-α/β) antiviral response. SARS-CoV-2 NSP3 expression suppressed poly(I:C)-induced IRF3 phosphorylation and broadly reduced cellular phosphorylation levels in a dose-dependent manner. These findings suggest that SARS-CoV-2 NSP3 modulates host phosphorylation dynamics to subvert antiviral signaling and facilitate viral replication.

**IMPORTANCE** Understanding host-virus and virus-virus interactions is essential for uncovering mechanisms of viral replication and immune evasion, and for identifying targets for rational antiviral intervention. While previous screens using individually expressed severe acute respiratory syndrome coronavirus 2 (SARS-CoV-2) proteins have revealed host factors involved in infection, they could not capture virus-virus protein interactions or virus-host interactions in the context of infection. Here, we engineered a recombinant (r)SARS-CoV-2 expressing a Strep-tagged nucleocapsid (N) protein to identify viral protein interactions with SARS-CoV-2 during active infection. We identified an interaction between the viral N protein and the nonstructural protein 3 (NSP3) and uncovered a previously unrecognized role for NSP3 in regulating viral and host protein phosphorylation, interaction with interferon regulatory factor 3 (IRF3), and regulation of innate immune response. This work highlights a powerful approach for dissecting protein interaction networks occurring during SARS-CoV-2 infection and suggests new targets for therapeutic development against SARS-CoV-2.

**KEYWORDS** SARS-CoV-2, NSP3, nucleocapsid protein, Strep-tag, IRF3

Address correspondence to Luis Martinez-Sobrido, lmartinez@txbiomed.org, or Chengjin Ye, cye@txbiomed.org.

L.M.-S. and C.Y. are co-inventors on a patent related to reverse genetics methods for generating recombinant SARS-CoV-2. The other authors do not report any conflict of interest.

See the funding table on p. 14.

Severe acute respiratory syndrome coronavirus 2 (SARS-CoV-2) is a highly transmissible and pathogenic coronavirus that emerged in December 2019 and caused an

acute respiratory disease devastating pandemic known as coronavirus disease 2019 (COVID-19) with profound impact on human health (1–3). SARS-CoV-2 is a positive-sense single-stranded RNA virus belonging to the *Coronaviridae* family, *Betacoronavirus* genus, which also includes SARS-CoV, Middle East respiratory syndrome-related coronavirus (MERS-CoV), and human coronavirus HKU1 and OC43 (4, 5). The SARS-CoV-2 genome is approximately 30 kilobases in length and encodes four structural proteins (spike [S], envelope [E], membrane [M], and nucleocapsid [N]), several accessory open reading frame (ORF) proteins (3a, 6, 7a, 7b, 8, 9b, and 10), and 16 nonstructural proteins (NSP1–NSP16) (6).

The nucleocapsid (N) protein of SARS-CoV-2 is a 46-kDa multifunctional RNA-binding protein essential to multiple stages of the viral life cycle. N encapsidates the viral genome into ribonucleoprotein (vRNP) complexes and facilitates their incorporation into virions through interaction with the M protein (7, 8). Structurally, N comprises an N-terminal domain (NTD), an RNA-binding domain (RBD), a central serine/arginine-rich intrinsically disordered region (SR-IDR or Linker domain), a dimerization domain (DD), and a C-terminal domain (CTD) (9, 10). SARS-CoV-2 N protein is highly phosphorylated during viral infection, particularly within the SR-IDR, and localizes to replication-transcription complexes (RTCs) in the early stages of infection, where it facilitates RNA synthesis and template switching (11, 12).

NSP3 is the largest coronavirus protein, with a molecular mass of ~200 kDa, and contains at least eight domains (13). These domains include the ubiquitin-like domain 1 (Ubl1), ADP-ribose phosphatase domain (ADRP), a SARS-unique domain (SUD), the ubiquitin-like domain 2 (Ubl2), the papain-like protease 2 (PLP2) domain, the nucleic-acid binding (NAB) domain, and domains Y1 and CoV-Y of unknown function, as well as two transmembrane (TM) regions (13). The ADRP is highly conserved across diverse coronaviruses and is involved in ADP-ribose binding (14–17). The PLP2 domain recognizes and cleaves LXGGX motifs in target proteins and removes ubiquitin and ISG15 modifications, thus counteracting innate immune signaling (18–21). Notably, SARS-CoV NSP3 ADRP has been previously suggested to have a phosphatase function (22).

The innate immune system plays a central role in recognizing and limiting viral infections (23). During coronavirus infection, viral RNA is detected by the cytosolic sensors, retinoic acid-inducible gene I (RIG-I) and/or melanoma differentiation gene 5 (MDA5), which activate the mitochondrial antiviral signaling protein (MAVS). This leads to downstream signaling through multiple kinases that phosphorylate interferon regulatory factor 3 (IRF3), promoting its dimerization and nuclear translocation to induce type I interferon (IFN-α/β) production (23–26). IFN-α/β then engage with their cognate receptor to activate the JAK-STAT signaling pathway, resulting in phosphorylation and nuclear translocation of STAT1 and STAT2. These transcription factors, together with IRF9, translocate to the nucleus to drive the expression of interferon-stimulated genes (ISGs), which collectively establish an antiviral state (26, 27). To establish infection, SARS-CoV-2 encodes multiple proteins that antagonize the IFN-α/β responses. These include NSP1 (28), NSP3 (29), NSP5 (30), NSP7 (31), NSP12 (32), NSP15 (33), ORF3b (34), M (35), ORF6 (36), and N (37, 38), which disrupt IFN-α/β production or downstream signaling, either directly or through interaction with host immune factors (39, 40).

In this study, we engineered a recombinant (r)SARS-CoV-2 expressing a Strep-tagged N protein to identify host and viral proteins involved in interaction with SARS-CoV-2 N protein during infection. Our results identified NSP3 as a viral interactor of N protein in infected cells. Importantly, we discovered a novel role for SARS-CoV-2 NSP3 in regulating the phosphorylation of N and cellular proteins, suggesting a novel mechanism by which SARS-CoV-2 modulates host innate immune responses. Collectively, these findings provide new insights into the interactome of SARS-CoV-2 N with viral and host proteins during viral infection, a new functional role of SARS-CoV-2 NSP3, and a powerful approach to identify viral and cellular proteins interacting with SARS-CoV-2 in the context of viral infection. Results from these studies can be used to identify new viral-

and/or host-directed therapeutics for the treatment of SARS-CoV-2 and other coronavirus infections.

## RESULTS

### Generation of a rSARS-CoV-2 expressing a Strep-tagged N

To investigate protein-protein interactions (PPIs) of SARS-CoV-2 N protein in the context of viral infection, we utilized a bacterial artificial chromosome (BAC)-based reverse genetics system (41) to engineer a Strep-tag at the N-terminal of SARS-CoV-2 N. A monomeric (m)Cherry reporter gene was inserted to facilitate tracking of viral infection (Fig. 1A). The recombinant virus, termed rSARS-CoV-2 mCherry-SN, was successfully rescued following BAC transfection into Vero E6 cells, as previously described (41). Immunofluorescence using an anti-Strep antibody showed co-localization with N protein in rSARS-CoV-2 mCherry-SN-infected cells (Fig. 1B). Moreover, a band corresponding to Strep-N was readily detected using the same Strep antibody by Western blot (Fig. 1C). Using an N-specific antibody, Western blot analysis further confirmed the expression of Strep-N, which has a higher molecular weight than wild-type (WT) N due to presence of the Strep tag (Fig. 1C). Importantly, both the mCherry reporter and the Strep-tag remained intact in the viral genome after nine serial passages in Vero E6 cells (Fig. S1). Sanger sequencing of the RT-PCR-amplified region between ORF8 and N in passage 9 verified the genome stability of rSARS-CoV-2 mCherry-SN (Fig. 1D). These results confirm the successful generation of a genetically stable rSARS-CoV-2 expressing Strep-tagged N, enabling the identification of N-interacting proteins during viral infection through Strep affinity-based methods.

### Identification of N interactions in SARS-CoV-2-infected cells

To identify N-interacting proteins, we infected A549-hACE2 cells with rSARS-CoV-2 mCherry-SN and performed Strep-mediated affinity purification followed by mass spectrometry (Fig. 2A). Cells infected with a control virus expressing untagged Strep N (rSARS-CoV-2 mCherry-N) were used as a negative control. Silver staining revealed the presence of multiple bands, with two prominent N-specific bands uniquely present in the rSARS-CoV-2 mCherry-SN pulldown (Fig. 2B; Fig. S2A). As a quality control, Ras GTPase-activating protein-binding protein 1 (G3BP1), a known SARS-CoV-2 N interactor, was successfully detected in the pulldown (Fig. S2B).

Mass spectrometry analysis identified multiple peptides derived from SARS-CoV-2 NSP3 (Fig. 2C), together with peptides corresponding to NSP4, NSP8, S, and M (Fig. S3). Immunoblotting of pulldown samples confirmed the presence of NSP3 in the pulldown using an NSP3-specific antibody (Fig. 2D). Co-immunoprecipitation in HEK293T cells co-transfected with NSP3-GFP and SARS-CoV-2 N fused to an HA epitope tag (N-HA) further confirmed the interaction between the two viral proteins in transfected cells (Fig. 2E). In addition, confocal microscopy of A549-hACE2 cells co-transfected with NSP3-GFP and N-mCherry showed that N formed cytoplasmic granule-like structures that co-localized with NSP3 (Fig. 2F). These data demonstrate the interaction of SARS-CoV-2 N and NSP3 in both infected and transfected cells.

### NSP3 interacts with N, via Ub1 and ADRP/Mac1 domains

To define the N region(s) responsible for interacting with NSP3, we generated N mutants lacking individual domains (Fig. 3A). Co-immunoprecipitation experiments revealed that deletion of the dimerization domain (dDD) abolished interaction of SARS-CoV-2 N with NSP3, while deletion of other domains (NTD, RBD, Linker, or CTD) did not affect the interaction of SARS-CoV-2 N with NSP3 (Fig. 3B). Notably, deletion of the NTD enhanced the interaction of N with NSP3, suggesting a negative regulatory role for the NTD. Consistently, fluorescent microscopy showed that N WT and dRBD, dLinker, and dCTD mutants formed cytoplasmic granules co-localizing with NSP3, whereas the N dDD

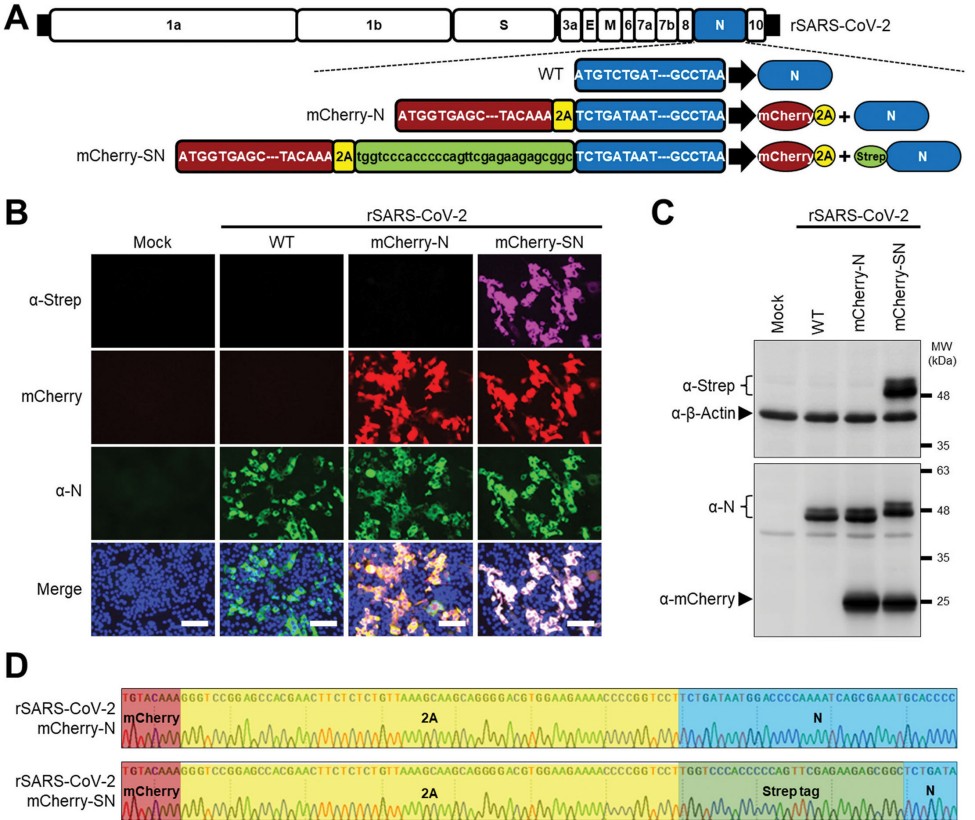

**FIG 1** Generation of a rSARS-CoV-2 expressing a Strep-tagged N. (A) Schematic diagram of the wild-type (WT), mCherry-N, and mCherry-SN rSARS-CoV-2 constructs. Representation of the viral proteins in the schematic representation of the viral genome is not drawn to scale. (B) Detection of Strep-N in Vero E6 cells infected with rSARS-CoV-2 mCherry-SN. Scale bar = 100 µm. (C) Western blot analysis of Vero E6 cell extracts mock-infected or infected (MOI = 1) with the indicated rSARS-CoV-2 after 48 h of infection. (D) Sanger sequencing confirmation of the Strep-tag insertion in the genome of passage 9 (P9) rSARS-CoV-2 mCherry-SN.

mutant lost this co-localization with NSP3 and the formation of cytoplasmic granules (Fig. 3C).

To identify the N-interacting region within NSP3, we constructed GFP-tagged NSP3 WT and deletion mutants (Fig. 3D). As shown in the co-immunoprecipitation result, the NSP3 dUbl1-ADRP mutant lacking the 1-412 N-terminal Ubl1 and ADRP domains failed to interact with N, whereas an NSP3 mutant containing the 1-412 N-terminal Ubl1 and ADRP domains was able to interact with N (Fig. 3E). However, NSP3 mutants containing the Ubl1 or the ADRP domains alone exhibited a reduced, but still detectable, interaction with N (Fig. 3E). Using confocal microscopy of A549-hACE2 cells co-transfected with SARS-CoV-2 N and the NSP3 WT or mutants, the images show co-localization of WT, Ubl1, ADRP, and Ubl1-ADRP, but dUbl1-ADRP, with N protein in the cytoplasm of transfected cells (Fig. 3F). Taken together, these findings demonstrate that the interaction of SARS-CoV-2 N protein with NSP3 is mediated by the N dimerization domain and the N-terminal 1-412 amino acids of NSP3 containing the Ubl1 and ADRP domains, with the NTD of N acting as a negative regulator of this interaction.

## NTD protects N from NSP3-associated dephosphorylation

Protein phosphorylation, primarily on serine (Ser, S), threonine (Thr, T), and/or tyrosine (Tyr, Y) residues, plays a key role in regulating protein functions (42, 43). SARS-CoV-2 N was previously described to be phosphorylated in infected and transfected cells (44, 45). To determine the phosphorylation status of SARS-CoV-2 N, we purified N from infected

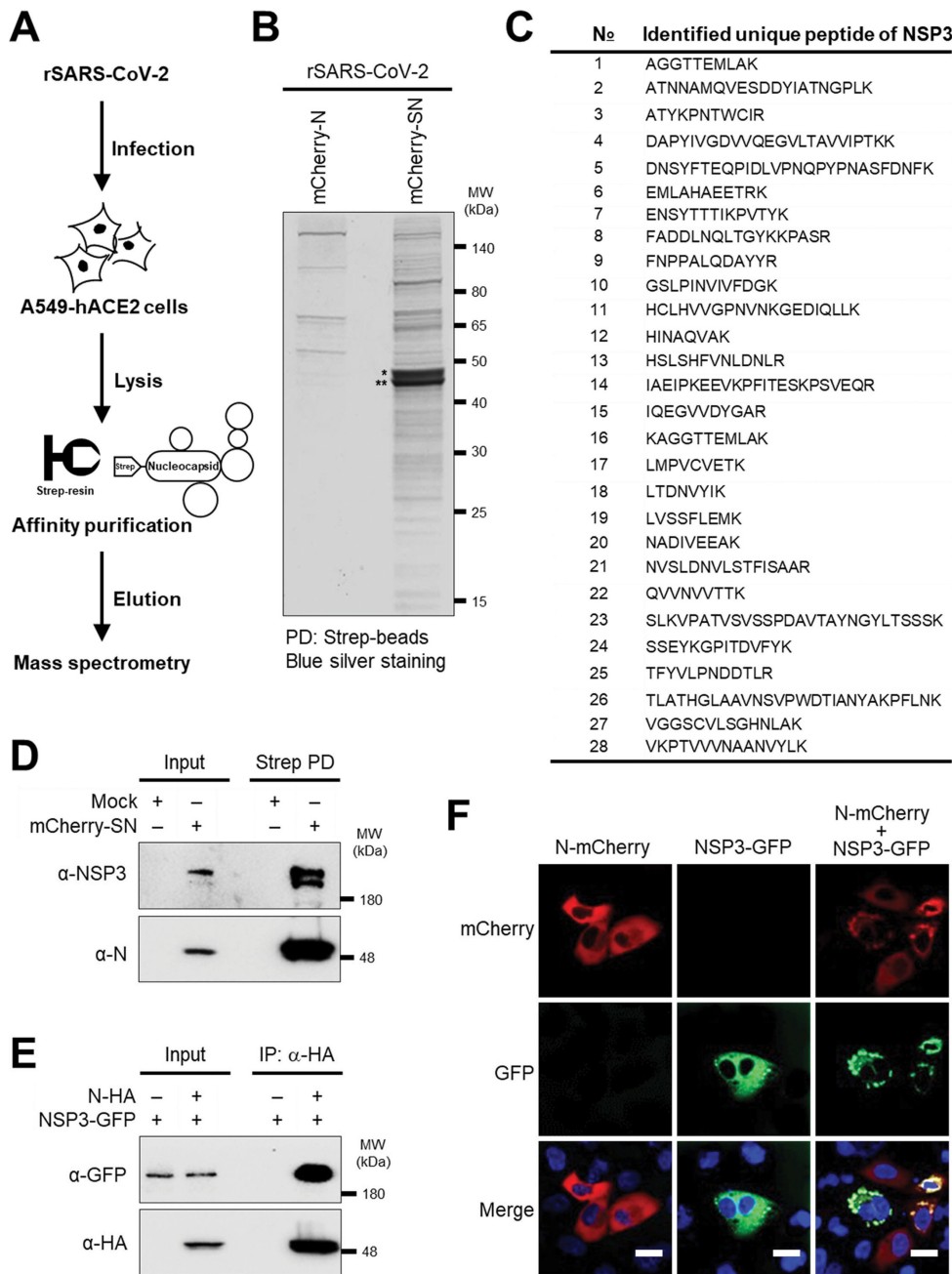

**FIG 2** Identification of N interactions in SARS-CoV-2-infected cells. (A) Experimental workflow for affinity purification-mass spectrometry (AP-MS). (B) Blue silver staining of eluted fractions after Strep affinity purification of N from A549-hACE2 cells infected with the indicated viruses. * Phosphorylated N; ** Unphosphorylated N. (C) Unique NSP3 peptides identified by AP-MS. (D) Co-immunoprecipitation of NSP3-N in SARS-CoV-2-infected A549-hACE2 cells. (E) Co-immunoprecipitation of N and NSP3 in transfected HEK293T cells. (F) Co-localization of NSP3 (green) and N (red) in transfected A549-hACE2 cells. Scale bar = 20 µm.

cells and probed with phospho-specific antibodies. N was recognized by anti-phospho-Ser antibodies (Fig. 4A), indicating serine-specific phosphorylation. This finding was further confirmed by ectopic expression of N in HEK293T transfected cells (Fig. 4B). Previous structural studies on SARS-CoV NSP3 suggested the presence of a functional phosphatase domain (22). To assess the impact of NSP3 on N phosphorylation, we co-transfected HEK293T cells with NSP3 together with N WT or N dNTD. While N WT phosphorylation remained unchanged, N-dNTD phosphorylation was markedly reduced

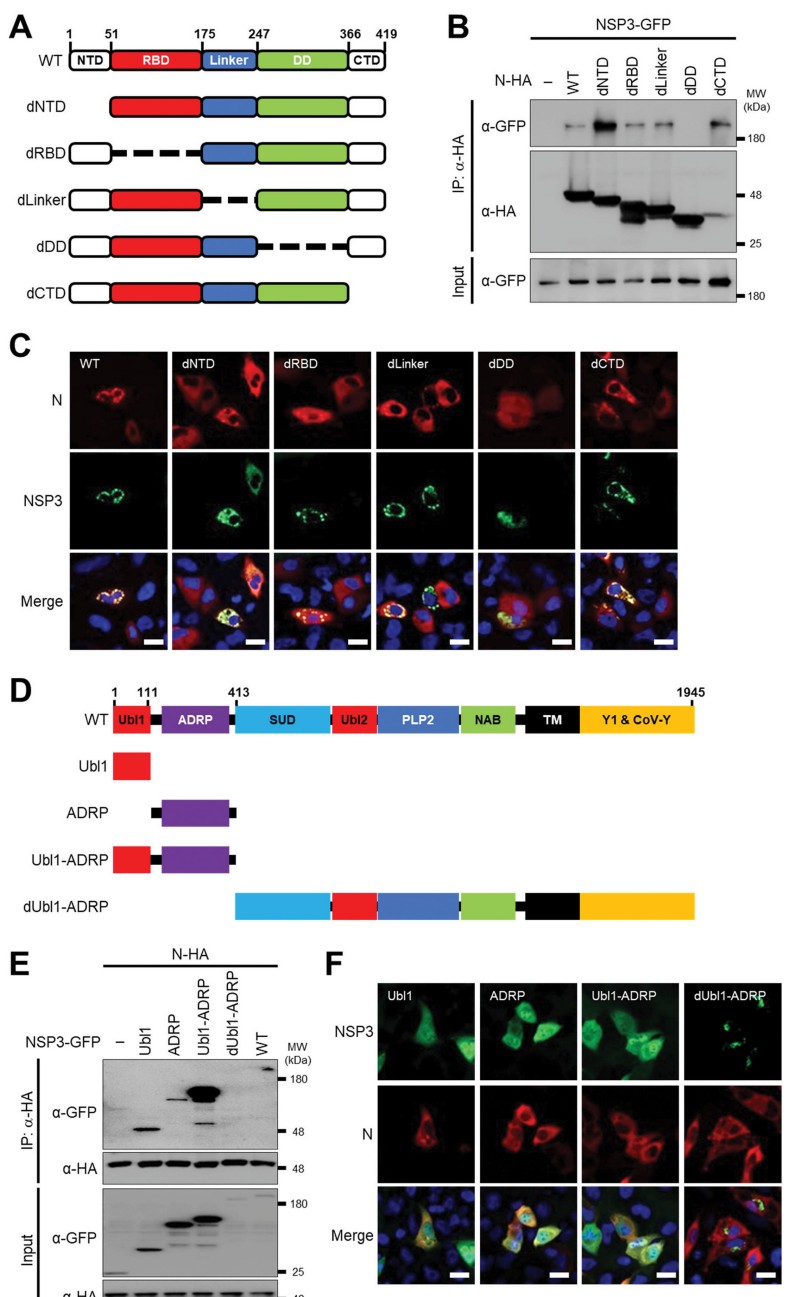

**FIG 3** NSP3 interacts with N, via Ub1 and ADRP/Mac1 domains. (A) Schematic representation of SARS-CoV-2 N domains and deletion mutants. (B) Co-immunoprecipitation of NSP3 with N WT and deletion mutants from transfected HEK293T cells. (C) Co-localization of NSP3 (green) with N WT and deletion mutants (red) in transfected A549-hACE2 cells. Scale bar = 20 μm. (D) Schematic diagram of SARS-CoV-2 NSP3 domains and truncation mutants. (E) Co-immunoprecipitation of NSP3 and its truncation mutants with N WT from transfected HEK293T cells. (F) Co-localization of N (red) with NSP3 WT and truncation mutants (green) in transfected A549-hACE2 cells. Scale bar = 20 μm.

in the presence of NSP3 (Fig. 4C), suggesting the NTD protects N from NSP3-associated dephosphorylation. Further experiments showed that only NSP3 WT and the Ubl1-ADRP mutant, but not other NSP3 deletion constructs, induced a shift in N-dNTD migration (Fig. 4D). Importantly, a fraction of the N-dNTD shifted downward, forming a distinct band that co-migrated with the dephosphorylated N-dNTD produced by NSP3 Ubl1-ADRP (Fig. 4E). These results demonstrate that SARS-CoV-2 N protein is phosphorylated

exclusively on serine residues, with NSP3 reducing phosphorylation of N-dNTD. At the same time, the N NTD prevents this NSP3-associated dephosphorylation, indicating that interaction with NSP3 is required to reduce the phosphorylation status.

## NSP3 inhibits IRF3 to suppress innate immune signaling

SARS-CoV-2 employs various strategies to suppress host immune responses (39, 40). NSP3 acts as a crucial virulence factor, partly by inhibiting IFN-α/β production (21, 46). Given that IRF3 phosphorylation is essential for type I IFN induction, we evaluated whether SARS-CoV-2 NSP3 affects IRF3 phosphorylation and activation. HEK293T cells expressing NSP3-GFP or empty-GFP were stimulated with poly(I:C), and IRF3 phosphorylation was assessed. Poly(I:C)-induced IRF3 phosphorylation was markedly suppressed by NSP3 expression (Fig. 5A). To further assess functional consequences, we measured IFN-β promoter activity using a luciferase reporter assay. Both poly(I:C) treatment and Sendai virus (SeV) infection robustly activated the IFN-β promoter, but this activation was inhibited in the presence of SARS-CoV-2 NSP3 (Fig. 5B and C, respectively). Consequently, interferon-stimulated response element (ISRE)-driven luciferase activity was also suppressed by NSP3 upon poly(I:C) treatment (Fig. 5D) or SeV infection (Fig. 5E). Finally, we evaluated the interaction of SARS-CoV-2 NSP3 with IRF3. To that end, we transfected NSP3-GFP together with FLAG-IRF3, or FLAG-empty, into HEK293T cells. NSP3-GFP was co-immunoprecipitated with FLAG-IRF3 (Fig. 5F), confirming the interaction between SARS-CoV-2 NSP3 and IRF3. Protein phosphorylation is a post-translational modification that plays important roles in several cell biological processes, including signal transduction (47, 48). To investigate the activity of NSP3 on phosphorylation of host proteins, we examined cell protein phosphorylation in the presence of SARS-CoV-2 NSP3. We probed cell lysates expressing increasing amounts of NSP3 for phospho-Ser, -Thr, and -Tyr residues. A dose-dependent reduction in all three types of phosphorylation levels was observed (Fig. 5G). Blots using ubiquitination K63 and ubiquitination WT (Fig. 5G) were included as negative and positive controls, respectively. Altogether, these data support a mechanism by which SARS-CoV-2 NSP3 interferes with phosphorylation of host proteins, including IRF3, to facilitate immune evasion and viral replication, suggesting the presence of a functional phosphatase domain, as previously suggested for SARS-CoV NSP3 (22).

## DISCUSSION

The SARS-CoV-2 genome encodes an array of structural and non-structural proteins that orchestrate complex interaction networks essential for viral replication and host manipulation (1, 6, 49–52). Understanding these interactions is key for developing novel antiviral strategies. Prior interactome studies using plasmid-transfected HEK-293T/17 cells identified host proteins interacting with individual SARS-CoV-2 proteins (50). However, such plasmid overexpression systems do not fully reflect the complexity of natural viral infections and fail to capture interactions between viral proteins. To address these limitations, we engineered an rSARS-CoV-2 expressing Strep-tagged N and mCherry separated by a P2A self-cleaving peptide derived from porcine teschovirus-1 (53, 54). In rSARS-CoV-2 mCherry-SN-infected cells, Strep-tagged N was readily detectable, and multiple protein bands were observed in Strep pulldown assays.

Using an affinity purification-mass spectrometry approach combined with rSARS-CoV-2 mCherry-SN infection, we identified five viral proteins that interact with SARS-CoV-2 N in the context of viral infection: S, M, NSP8, NSP4, and NSP3. The S protein, a trimeric structure on the viral envelope, mediates receptor binding and viral entry (55). Interactions between N and S have been reported during coronavirus infections, including MERS-CoV and SARS-CoV-2, and disrupting this interaction inhibits viral replication (56). Similarly, the interaction between N and M, also identified in our study, is essential for virion assembly in MHV (7, 57) and SARS-CoV (58, 59). This confirms the feasibility of identifying both viral proteins that interact with SARS-CoV-2 N in the context of viral infection. Furthermore, our results demonstrate the feasibility of using similar

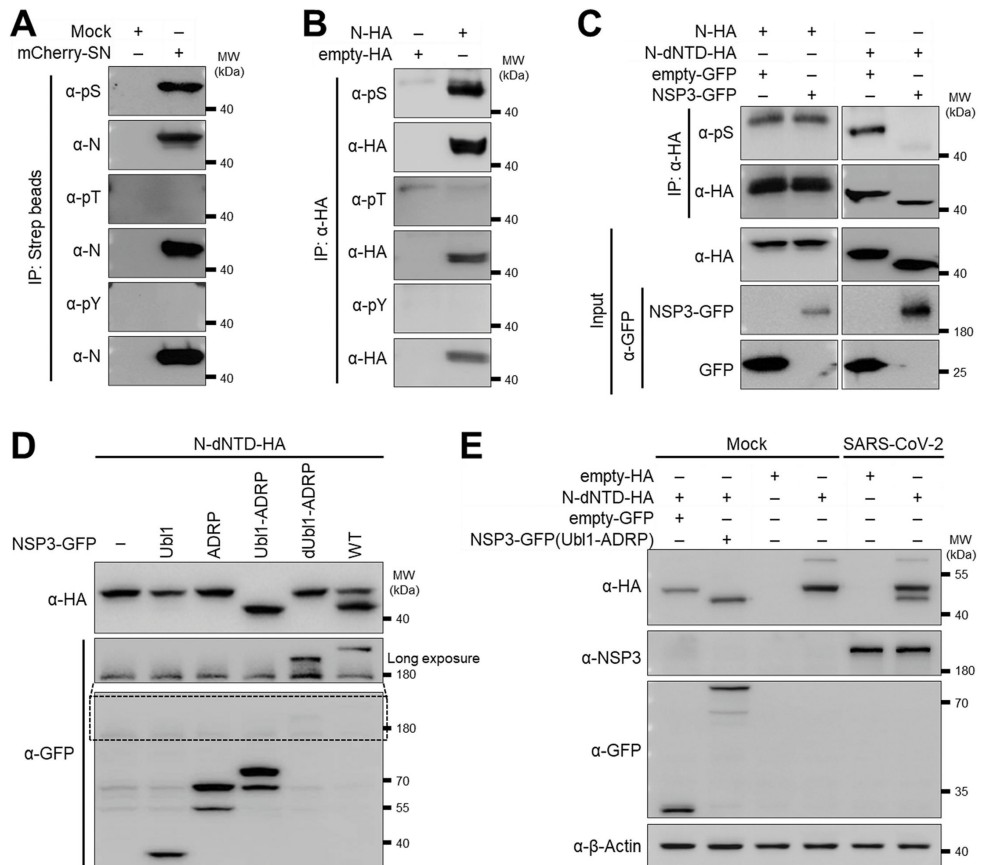

**FIG 4** NTD protects N from NSP3-associated dephosphorylation. (A) Western blot analysis of N phosphorylation in SARS-CoV-2-infected A549-hACE2 cells. (B) Western blot of N phosphorylation in HEK293T cells transfected with N expression plasmids. (C) Comparison of phosphorylation levels of N and N-dNTD in the presence or absence of NSP3 in transfected HEK293T cells. (D) Western blot analysis showing the effect of NSP3 truncation mutants on N-dNTD phosphorylation in transfected HEK293T cells. (E) Western blot analysis showing the effect of SARS-CoV-2 infection on N-dNTD phosphorylation in A549-ACE2 cells.

recombinant viruses carrying Strep-tagged proteins to identify viral and/or cellular proteins interacting with other viral proteins in the context of SARS-CoV-2 infection.

Phosphorylation is a critical post-translational modification for the functions of both viral and host proteins (47). SARS-CoV-2 N, a serine/arginine SR-rich protein, is known to be regulated by phosphorylation (45, 60, 61), particularly within conserved domains essential for replication (62, 63). Our findings reveal that SARS-CoV-2 N-dNTD is dephosphorylated by NSP3, a process mediated by the interaction of N with NSP3, which has been observed in other coronaviruses (7, 64–67). Specifically, SARS-CoV-2 N is phosphorylated exclusively on serine residues in both infected and transfected cells. Interestingly, while full-length N retains its phosphorylation status in the presence of NSP3, deletion of SARS-CoV-2 N protein NTD renders it susceptible to dephosphorylation by NSP3 or the Ubl1-ADRP domain. This suggests that the SARS-CoV-2 N protein NTD shields N from NSP3-associated dephosphorylation, thereby maintaining N phosphorylation in a dynamic balance. Targeting N phosphorylation has previously shown therapeutic promise. For example, SRPK inhibitors reduce N phosphorylation and inhibit SARS-CoV-2 replication *in vitro* (44). Our study demonstrates that the SARS-CoV-2 N protein NTD prevents NSP3-associated dephosphorylation, whereas NSP3 directly impairs IRF3 phosphorylation and subsequently disrupts innate antiviral signaling. These dual activities of SARS-CoV-2 NSP3 underscore its central role in viral replication and immune evasion.

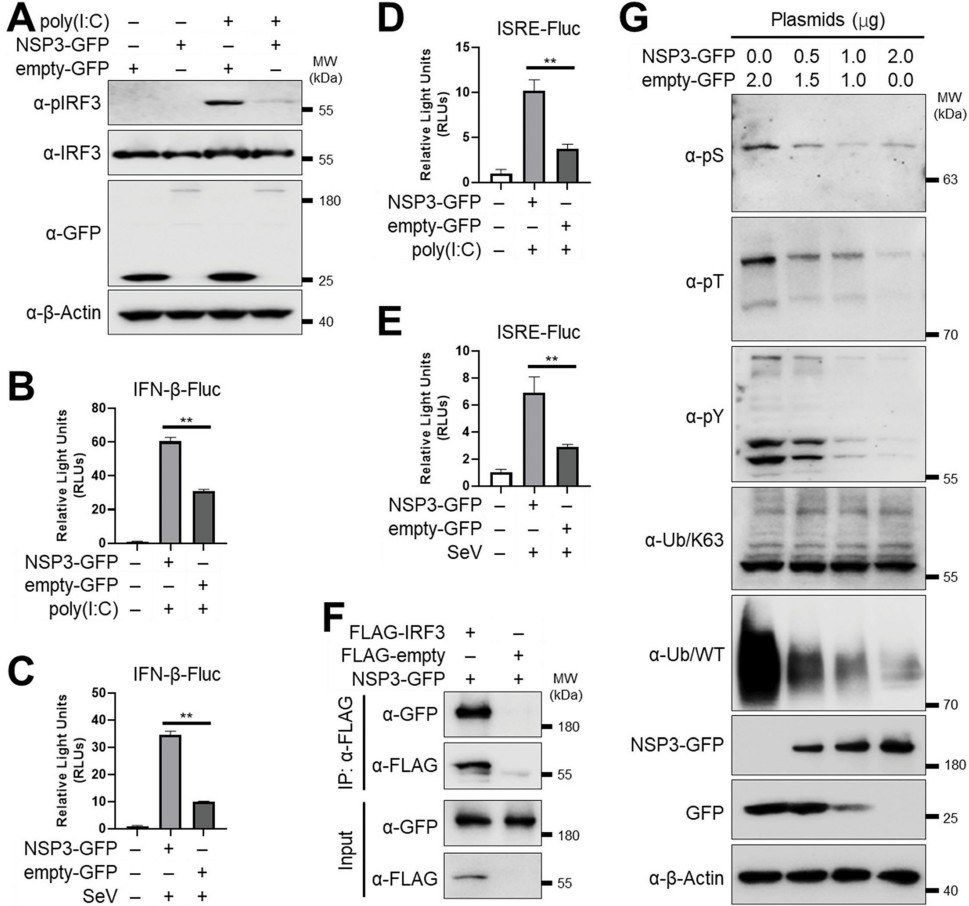

**FIG 5** NSP3 inhibits IRF3 to suppress innate immune signaling. (A) Western blot analysis of IRF3 phosphorylation in HEK293T cells transfected with NSP3. (B and C) NSP3-associated inhibition of poly(I:C)-induced (B) and SeV-induced (C) IFN-β promoter activation in transfected HEK293T cells. (D and E) NSP3 inhibition of poly(I:C)-induced (D) and SeV-induced (E) ISRE promoter activation in transfected HEK293T cells. (F) Co-immunoprecipitation of NSP3 and IRF3 in HEK293T transfected cells. (G) Western blot analysis of global protein phosphorylation in HEK293T cells transfected with increasing amounts of NSP3-expressing plasmid. \*\*$P < 0.01$.

NSP3, the largest non-structural protein of SARS-CoV-2, plays a central role in viral replication and transcription (13). Together with NSP4, it induces the formation of double-membrane vesicles (DMVs), a hallmark of coronavirus replication (68–70). Importantly, recent studies have independently validated the interaction of SARS-CoV-2 N with NSP3, including its critical role in viral pathogenicity (71). However, our study reveals a novel function of NSP3 in suppressing innate immunity through its interaction with IRF3, thereby inhibiting IRF3 phosphorylation, a prerequisite for IRF3 nuclear translocation and the activation of IFN and ISRE promoters (72). NSP3 also exhibits deubiquitinase and deISGylase activities, structurally resembling human deubiquitinases (13, 17, 18, 20, 73). These activities help antagonize host post-translational modifications, such as ubiquitylation and ISGylation, which are crucial for antiviral signaling (74–76). For example, NSP3 contains a well-characterized deISGylase function within its PLpro domain, enabling it to remove ISG15 modifications from host proteins such as MDA5, thereby facilitating viral evasion of antiviral defenses by disrupting ISG15 conjugation on MDA5 (77). However, the dephosphorylation of N-dNTD observed in this study differs from the N ISGylation modification reported in previous research, as the dephosphorylated N-dNTD exhibits a size difference of less than 15 kDa, unlike the larger shift seen between N and ISGylated N (76).

A recent study demonstrated that SARS-CoV-2 mutants lacking the NSP3 MAC1 domain show reduced IFN antagonism and attenuated pathogenicity *in vivo* (78). Our data suggest that this attenuation may be linked to NSP3's role in IRF3 dephosphorylation. Furthermore, we observed that SARS-CoV-2 NSP3 alters global protein phosphorylation in infected cells, raising the possibility that it harbors a functional phosphatase domain—a hypothesis supported by earlier structural studies on SARS-CoV NSP3 (22). Additional work is needed to fully elucidate the potential phosphatase activity of SARS-CoV-2 NSP3 and its implications during viral infection. To facilitate the identification of other cellular proteins targeted by NSP3, a similar Strep-tag in SARS-CoV-2 NSP3 like the one described in this manuscript for the N protein may be a promising option. In addition, it is worth noting that the Macro domain of NSP3 in SARS-CoV-2 and other coronaviruses is essential for viral replication and has led several groups to develop inhibitors targeting the NSP3 Macro domain using *in vitro* (79) and crystallographic (80, 81) screenings.

In summary, we propose a working model wherein SARS-CoV-2 N protein NTD protects N from NSP3-induced dephosphorylation. In contrast, the N protein DD mediates interaction with NSP3 by interacting with the N-terminal domain of NSP3 (Fig. 6). Separately, SARS-CoV-2 NSP3 inhibits IRF3 activation, suppressing IRF3-mediated host antiviral responses. Finally, our strategy using rSARS-CoV-2 expressing Strep-tagged proteins offers a powerful and adaptable platform for identifying both viral and host protein interactors under natural infection conditions, paving the way to identify virus-virus and virus-host interactions for the rational design of novel therapeutics targeting critical interactions in the context of viral infection.

## MATERIALS AND METHODS

### Biosafety

All the experiments with infectious SARS-CoV-2 were conducted under appropriate biosafety level 3 (BSL3) laboratories at the Texas Biomedical Research Institute (Texas Biomed) and were approved by the Texas Biomed Institutional Biosafety Committee.

### Cells

Human embryonic kidney (HEK) 293T (CRL-11268) and African green monkey kidney epithelial (Vero E6, CRL-1586) cells were obtained from the American Type Culture Collection. The human lung carcinoma (A549) cell line stably expressing human ACE2 (A549-hACE2, NR-53821) was obtained from BEI Resources. All cell lines were maintained in Dulbecco's modified Eagle medium (DMEM) supplemented with 10% (vol/vol) fetal bovine serum (FBS, VWR) and 1% PSG (100 U/mL penicillin, 100 µg/mL Streptomycin, and 2 mM L-glutamine; Corning), and routinely cultured at 37°C with 5% $CO_2$.

### Viruses

The recombinant (r)SARS-CoV-2 WT and rSARS-CoV-2 mCherry-N were previously described (41, 54). The rSARS-CoV-2 mCherry-SN expressing a Strep-tagged N was rescued using the previously described BAC-based SARS-CoV-2 reverse genetics system (41). The BAC containing the entire SARS-CoV-2 WA1 sequence was genetically engineered to express mCherry-2A-Strep fused to the N protein N terminus using standard molecular biology techniques (82). The resultant BAC was transfected into monolayers of Vero E6 cells using Lipofectamine 2000 according to the manufacturer's instructions for virus rescue. At 24 h post-transfection, the cell culture supernatant was replaced with post-infection media (DMEM containing 2% FBS and 1% PSG), and the culture was collected as passage 0 (P0). A P1 stock was generated by infecting fresh Vero E6 cells with P0 at a multiplicity of infection (MOI) of 0.0001 for further use.

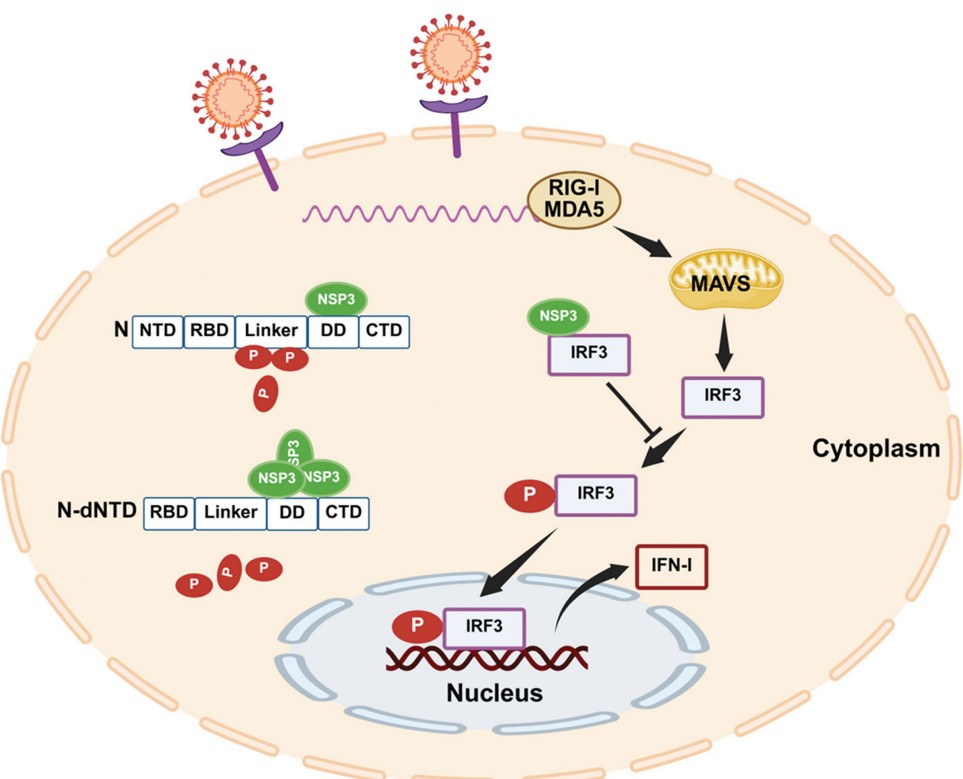

**FIG 6** Proposed model of NSP3-N interactions and functions during SARS-CoV-2 infection. NSP3 interacts with N via its N-terminal domain and dephosphorylates N through binding to the N dimerization domain (DD), while the N protein N-terminal domain (NTD) shields it from dephosphorylation. NSP3 also binds to IRF3, inhibiting its phosphorylation and thereby suppressing SARS-CoV-2 infection-induced type I interferon (IFN-I) responses. Illustration created using BioRender.

## Antibodies and other reagents

The mouse anti-SARS-CoV N monoclonal antibody (mAb) 1C7C7 cross-reacting with SARS-CoV-2 N protein was a gift from Thomas Moran at The Icahn School of Medicine at Mount Sinai. Anti-Flag M2 affinity agarose (A2220), EZview Red anti-HA affinity gel (E6779), 4′,6-diamidino-2-phenylindole dihydrochloride (DAPI) (D9542), polyinosinic-polycytidylic acid sodium salt (polyI:C) (P0913), mouse anti-phosphoserine mAb, clone 4A4 (05-1000), mouse anti-phospho-Threonine mAb (05-1923), mouse anti-β-Actin mAb (A1978), mouse anti-HA tag mAb (H9658), mouse anti-FLAG M2 mAb (F1804), and mouse anti-GFP mAb (G1546) were purchased from Sigma-Aldrich. Rabbit anti-IRF3 mAb (11904), rabbit anti-pIRF3 mAb (4947), rabbit anti-K63-linkage specific polyubiquitin mAb (5621), mouse anti-Phospho-Tyrosine (4G10) mAb (96215), rabbit anti-SARS-CoV-2 NSP3 polyclonal antibody (pAb) (88086), and rabbit anti-G3BP1 pAb (17798) were purchased from Cell Signaling Technology. Rabbit anti-mCherry pAb (168-11081) and rabbit anti-Strep tag pAb (A00626) were purchased from Raybiotech and GenScript, respectively. Sheep anti-mouse IgG horseradish peroxidase (HRP)-conjugated secondary antibody (NA931) and donkey anti-rabbit IgG HRP-conjugated secondary antibody (NA934) were obtained from Cytiva. Donkey anti-rabbit IgG fluorescein (FITC)-conjugated secondary antibody (111-095-003) was purchased from Jackson ImmunoResearch Laboratories. Dual-Luciferase Reporter Assay System (E2980) and Strep-TactinXT beads (2-5010-002) were purchased from Promega and IBA Lifesciences, respectively. ABC-HRP kit (PK-6102) and DAB substrate kit (SK-4100) were purchased from Vector Laboratories.

## Plaque assays

Monolayers of Vero E6 or A549-hACE2 cells (six-well plate format, duplicates) were infected with 10-fold serial dilutions of each virus for 1 h at 37°C. After viral adsorption, cells were washed three times with phosphate-buffered saline (PBS), covered with post-infection media containing 1% low-melting agar, and returned to the 37°C 5% $CO_2$ incubator. At 72 h post-infection, the cells were fixed with 10% formalin overnight, and then the plates were moved to the BSL2 laboratory. In the BSL2 laboratory, the agar was removed, and the plates were imaged for mCherry expression under a ChemiDoc MP Imaging System (Bio-Rad). Then, the cells were subsequently incubated with the Strep antibody and the goat anti-rabbit IgG FITC-conjugated secondary antibody and imaged under a ChemiDoc MP Imaging System. Finally, the cells were stained with the SARS-CoV cross-reactive 1C7C7 monoclonal antibody and an ABC-HRP kit. Viral plaques were visualized by staining the cells with a DAB Substrate kit, and images were taken using a ChemiDoc MP Imaging System. The plaque assay was also used to determine virus titers in Vero E6 cells. The viral titers were calculated by counting plaques and expressed as PFU/mL.

## Reverse-transcription PCR (RT-PCR)

Confluent monolayers of Vero E6 cells (six-well plate format) were infected with rSARS-CoV-2 at an MOI 0.1. At 24 h post-infection, total RNA was extracted from cells using TRIzol (1 mL/well; Thermo Fisher Scientific) per the manufacturer's instructions. Reverse transcription and cDNA synthesis were performed with the First-Strand Synthesis System for RT-PCR (Thermo Fisher Scientific). The synthesized cDNA was used to amplify the region between ORF8 and N by PCR. The PCR products were gel-purified and subsequently subjected to Sanger sequencing. RT-PCR conditions and primer sequences are available upon request.

## Plasmids and transfections

Standard molecular biology procedures were used to generate the different plasmid constructs. The codon-optimized SARS-CoV-2 NSP3 gene was obtained from Addgene (#141257) and subcloned into pCAGGS-MCS-GFP. The full-length cDNA encoding IRF3 was obtained from Addgene (#32713) and subcloned into the pCMV-FLAG-MCS (Clontech). The N coding sequences were amplified from SARS-CoV-2-infected Vero E6 cells by RT-PCR and subcloned into pCAGGS-MCS-HA. All constructs were validated by Oxford Nanopore Technologies (Plasmidsaurus) and were transfected into cells using Lipofectamine 2000 (Thermo Fisher Scientific). Primer sequences used for cloning are available upon request.

## Western blot

Protein fractions were used to load SDS polyacrylamide gel electrophoresis (SDS-PAGE) gels to perform Western blots. Briefly, the cells were lysed in 1× passive lysis buffer (Promega) at 4°C for 1 h and then centrifuged at 18,000 × $g$ at 4°C for 15 min. A 10% SDS-PAGE was used to load equivalent amounts of cell lysates. Proteins were transferred to nitrocellulose membranes and blocked with 5% skimmed milk in PBS containing 0.1% Tween 20 at room temperature (RT) for 1 h. Afterward, membranes were incubated overnight with the indicated primary antibodies at 4°C, followed by HRP-conjugated secondary antibodies at 37°C for 1 h. Finally, membranes were developed using an ECL substrate (Thermo Fisher Scientific) under a ChemiDoc MP Imaging System. An antibody against β-Actin was used as a loading control.

## Affinity purification

Human A549-hACE2 cells in T75 flasks were infected with rSARS-CoV-2 mCherry-N or rSARS-CoV-2 mCherry-SN at an MOI of 0.1. At 48 h post-infection, when comparable

levels of viral infection were assessed by mCherry expression, cells were lysed with 1% NP-40 lysis buffer for 30 min at 4°C. The cell lysates were centrifuged at 18,000 × $g$ for 15 min at 4°C to remove cell debris. The clarified cell lysates were heat-inactivated according to the institutionally approved inactivation protocol and moved to BSL2 laboratories. Then, affinity purification was completed in the BSL2 laboratory at RT according to the manufacturer's instructions. Briefly, the inactivated cell lysates were added to the Strep-TactinXT beads column to flow through by gravity. The columns were washed five times with 1 mL of 1× Buffer W each time. The 1× Buffer BXT was used to elute proteins. The purified samples were analyzed using SDS-PAGE and blue silver staining.

## SDS-PAGE and mass spectrometry

SDS-PAGE, mass spectrometry, and proteomics data analysis were performed as previously described (83, 84). A small fraction of the eluted samples was boiled for 10 min in SDS sample buffer before electrophoresis on Criterion XT MOPS SDS-PAGE reducing gels (Bio-Rad), and protein bands were visualized by blue silver staining. The rest of the elution samples were digested *in situ* with trypsin (Promega) in 40 mM $NH_4HCO_3$ overnight at 37°C. The resulting tryptic peptides were analyzed by HPLC-ESI-tandem MS (HPLC-ESI-MS/MS) on a Thermo Fisher LTQ Orbitrap Velos mass spectrometer fitted with a New Objective Digital PicoView 550 NanoESI source. An Eksigent/AB Sciex NanoLC-Ultra 2-D HPLC system was used for the online HPLC separation of the digests. The column used was PicoFrit (New Objective; 75 µm i.d.), which was packed to 15 cm with C18 adsorbent (Vydac; 218MSB5 5 µm, 300 Å). The mobile phase A consisted of 0.5% acetic acid (HAc) and 0.005% trifluoroacetic acid (TFA), while the mobile phase B was made up of 90% ACN, 0.5% HAc, and 0.005% TFA. The gradient was set to 2–42% B over 120 min at a flow rate of 0.4 µL/min. The Orbitrap acquired precursor ions at a resolution of 60,000 (*m/z* 400) in centroid mode. Simultaneously, data-dependent collision-induced dissociation (CID) spectra of the 10 most intense ions in the precursor scan above a threshold of 3,000 were acquired in the linear trap (with an isolation window of 3 for MS/MS and a relative collision energy of 30). Ions with a 1+ or unassigned charge state were not fragmented. The dynamic exclusion settings were as follows: repeat count −1, repeat duration −30 s, exclusion list size −500, and exclusion duration −30 s. Scaffold 5 (v 5.2.1; Proteome Software) was utilized for proteomic data analysis. Peptide quality filters used for viewing and exporting proteomics results: 99.9% peptide, minimum of two peptides, and 99.9% protein. These settings resulted in a protein-level FDR of less than 1%.

## Co-immunoprecipitation

Co-immunoprecipitation assays were carried out using previously established protocols with some adjustments (85, 86). HEK293T cells were transfected with the indicated plasmids, lysed using 1% NP-40 lysis buffer at 36 h post-transfection, and clarified by centrifugation at 18,000 × $g$ for 15 min at 4°C. The clarified cell lysates were incubated with Protein G PLUS-Agarose for 1 h at RT to eliminate non-specific bindings. The supernatants were then precipitated by incubating with Strep- or HA-beads for 1 h at RT. Immunoprecipitated proteins were washed three times with 1% NP-40 lysis buffer and analyzed by Western blot with the indicated antibodies.

## Immunofluorescence assay

Monolayers of Vero E6 cells (six-well plate format, triplicate) were mock-inoculated or inoculated with the indicated viruses. At 48 h post-infection, the cells were fixed with 10% formalin overnight, and the plates were moved to the BSL2 laboratory by complying with the institutional biosafety policies. In the BSL2 laboratory, the cells were permeabilized using 0.5% (vol/vol) Triton X-100 in PBS for 15 min at RT, incubated with the indicated primary antibodies at 37°C for 1 h, washed with PBS, and stained with the corresponding secondary antibodies. Finally, the cells were imaged under an EVOS cell imaging system (Thermo Fisher Scientific).

## Confocal microscopy

Human A549-hACE2 cells were seeded on coverslips in six-well plates and transfected with the indicated plasmids. At 48 h post-transfection, the cells were fixed in 4% paraformaldehyde for 1 h at RT, permeabilized with PBS containing 0.5% Triton X-100, and stained with DAPI. Fluorescence images were acquired and analyzed using an LSM 880 confocal laser scanning microscope (Zeiss).

## Reporter gene assays

HEK293T cells (12-well plate format, triplicate) were transfected with the indicated Firefly luciferase (Fluc) reporter and indicated protein expression plasmids. A plasmid constitutively expressing Renilla luciferase (Rluc), pRL-TK, was included to normalize transfection efficiency. At 24 h post-transfection, transfected cells were either stimulated by poly(I:C) (1 µg/well) or SeV infection (100 HA units/well) for another 12 h. The cells were harvested, and the cell lysates were used to determine luciferase activities using a Dual-Luciferase Reporter Assay System. Fluc activity levels were normalized to Rluc activity levels as previously described (87).

## Statistical analysis

GraphPad Prism V8.0.1 was used for statistical analysis. Data are presented as mean ± standard deviation. Student's $t$-test determined differences between groups. $P < 0.05$ (\*) and $P < 0.01$ (\*\*) were considered significant.

## ACKNOWLEDGMENTS

This study was supported by the Emerging Respiratory Pathogens Award (ERP-1420664) from the American Lung Association (C.Y.) and Texas Biomed Forum Awards (L.M.-S. and C.Y.). H.Y. is a recipient of the Yangzhou University International Academic Exchange Fund.

We thank Susan T. Weintraub at the University of Texas Health Center at San Antonio for her assistance with the mass spectrometry analysis. We thank Fritz Roth and BEI Resources for providing the NSP3 plasmid and A549-hACE2 cells, respectively. We also thank Thomas Moran at The Icahn School of Medicine at Mount Sinai for providing the mouse anti-SARS-CoV-2 N monoclonal antibody 1C7C7.

## AUTHOR AFFILIATIONS

[1]Texas Biomedical Research Institute, San Antonio, Texas, USA
[2]College of Veterinary Medicine, Yangzhou University, Yangzhou, Jiangsu, China

## AUTHOR ORCIDs

Luis Martinez-Sobrido (iD) http://orcid.org/0000-0001-7084-0804
Chengjin Ye (iD) http://orcid.org/0000-0002-1934-9494

## FUNDING

| Funder | Grant(s) | Author(s) |
| --- | --- | --- |
| American Lung Association | ERP-1420664 | Chengjin Ye |
| Texas Biomedical Forum | 2025 | Chengjin Ye |
| Texas Biomedical Forum | 2023 | Luis Martinez-Sobrido |

## AUTHOR CONTRIBUTIONS

Hui Yang, Investigation, Writing – original draft | Daxin Peng, Formal analysis, Writing – review and editing | Luis Martinez-Sobrido, Conceptualization, Funding acquisition,

Supervision, Writing – review and editing | Chengjin Ye, Conceptualization, Formal analysis, Funding acquisition, Investigation, Methodology, Writing – original draft, Writing – review and editing

## ADDITIONAL FILES

The following material is available online.

### Supplemental Material

**Supplemental material (Spectrum02915-25-s0001.docx).** Viral sequences data; Fig. S1 to S3.

### Open Peer Review

**PEER REVIEW HISTORY (review-history.pdf).** An accounting of the reviewer comments and feedback.

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
