## [Reviewer comments · Microbiology Spectrum]

Microbiology Spectrum

SARS-CoV-2 Nonstructural Protein 3 Remodels the Phosphorylation of Target Proteins via Protein-Protein Interactions

Hui Yang, Daxin Peng, Luis Martinez-Sobrido, and Chengjin Ye

Corresponding Author(s): Chengjin Ye, Texas Biomedical Research Institute

Review Timeline:

Submission Date:	October 14, 2025
Editorial Decision:	January 21, 2026
Revision Received:	January 30, 2026
Accepted:	February 5, 2026

Editor: Rafael A. Medina

Reviewer(s): The reviewers have opted to remain anonymous.

Transaction Report:

DOI: <https://doi.org/10.1128/spectrum.02915-25>

Re: Spectrum02915-25 (**SARS-CoV-2 Nonstructural Protein 3 Remodels the Phosphorylation of Target Proteins via Protein-Protein Interactions**)

Dear Dr. Chengjin Ye:

Thank you for the privilege of reviewing your work. I am pleased to inform you that your manuscript has been editorially accepted for publication. However, there are a few additional questions in the submission form and a couple of comment form the reviews that need to be answered before the final decision. Once these are completed, please return your submission so that I can move your paper forward to acceptance.

Below you will find instructions from the Spectrum editorial office, and the reviewer comments.

Revision Guidelines

Sincerely,
Rafael A. Medina
Editor
Microbiology Spectrum

Reviewer #1 (Comments for the Author):

My comments from the previous review have been adequately addressed. The only modifications I would request are:

1. make available the sequence of the modified viruses via genbank or equivalent.

2. Revise language as it a bit loose in places. The ADRP domain is an ADP ribose 1' phosphatase, not a protein phosphatase as suggested by lines 35-36, 98-99, a section heading and a figure legend. Proteins do get poly-ADP-ribosylated, but you would need a different antibody than the anti-pS, anti-pT and anti-pY used in Fig. 4 to check for that. The bioinformatics of nsp3 has been reasonably well-trod by this point, and I don't think anyone has identified a putative protein phosphatase domain in SARS-CoV-2 nsp3 - if it's out there, please cite it. I'd recommend changing nsp3-mediated to nsp3-induced or nsp3-associated and removing the comments noted above.

Reviewer #1 (Comments for the Author): My comments from the previous review have been adequately addressed. The only modifications I would request are:

Comment 1. Make available the sequence of the modified viruses via genbank or equivalent.

Response: *We want to thank the reviewer for this valuable suggestion. Based on the comment made by the reviewer, we have included the full-length sequences of rSARS-CoV-2 mCherry-N and rSARS-CoV-2 mCherry-SN as a supplemental material in the revised manuscript.*

Comment 2. Revise language as it a bit loose in places. The ADRP domain is an ADP ribose 1' phosphatase, not a protein phosphatase as suggested by lines 35-36, 98-99, a section heading and a figure legend. Proteins do get poly-ADP-ribosylated, but you would need a different antibody than the anti-pS, anti-pT and anti-pY used in Fig. 4 to check for that. The bioinformatics of nsp3 has been reasonably well-trod by this point, and I don't think anyone has identified a putative protein phosphatase domain in SARS-CoV-2 nsp3 - if it's out there, please cite it. I'd recommend changing nsp3-mediated to nsp3-induced or nsp3-associated and removing the comments noted above.

Response: *We apologize for the confusion and thank the reviewer for this valuable comment. In lines 35-36 we propose that NSP3 may function as a potential viral phosphatase based on the findings of our study. However, we do not conclude that this potential phosphatase activity is attributable to the ADRP domain. In fact, the ADRP domain alone exhibited no phosphatase activity (Fig. 4E). The potential phosphatase activity appears to require the combined presence of the Ubl1 and ADRP domains, as neither Ubl1 nor ADRP alone showed phosphatase activity (Fig. 4E). The conclusion drawn from lines 98-99 was based on a study published in 2005 (Saikatendu KS et al., Structure, 2005). We cited this reference to provide a comprehensive overview of previous studies on NSP3. Following the recommendation made by the*

reviewer, all occurrences of “nsp3-mediated” have been revised to “nsp3-associated” in the revised manuscript (lines 179, 190, 196, 256, 260, 794, and 805).

Re: Spectrum02915-25R1 (**SARS-CoV-2 Nonstructural Protein 3 Remodels the Phosphorylation of Target Proteins via Protein-Protein Interactions**)

Dear Dr. Chengjin Ye:

We appreciate your consideration of the reviewers' comments and for the submission of a revised version.

I am pleased to inform you that your manuscript has been accepted, and I am forwarding it to the ASM production staff for publication. Your paper will first be checked to make sure all elements meet the technical requirements. ASM staff will contact you if anything needs to be revised before copyediting and production can begin. Otherwise, you will be notified when your proofs are ready to be viewed.

Sincerely,
Rafael A. Medina
Editor
Microbiology Spectrum